# Menstrual hygiene management practices after the chhaugoth demolition campaign in Achham, Nepal

Kalpana Jnawali[1,2], Prativa Tiwari[3], Ayusha Ghale[4*], Jagat Prasad Upadhyay[3], Anjana Sigdel[1], Sapana Thapa[1], Anushka Shrestha[4], K. C. Kiran[1], Sirjana Pandit Pahari[5], Damaru Prasad Paneru[3]

1 LA Grandee International College, Pokhara University, Pokhara, Nepal, 2 Nepal Red Cross Society, Kaski, Pokhara, Nepal, 3 School of Health and Allied Sciences, Pokhara University, Pokhara, Nepal, 4 Synergy Sphere Solutions Pvt. Ltd., Pokhara, Nepal, 5 Faculty of Health Science, Pokhara University, Pokhara, Nepal

* ghaleyaayusha@gmail.com

## Abstract

### Background

Menstruation, also called as *Chhau* (in Sudurpaschim and Karnali provinces of Nepal); is inextricably connected to social taboos and stigma in Nepal. Women who menstruate known as "*Chhaupadi*", are traditionally bound to stay in secluded huts called *Chhaugoth* during their menstrual periods and after child birth. Such practices have adverse social, economic and health consequences. Although *Chhaupadi* is legally punishable in Nepal; this practice remains prevalent. In 2017, the Government of Nepal launched the *Chhaugoth* demolition campaign to eliminate this practices. In this context, this study aimed to identify the menstrual hygiene management practices after the *Chhaugoth* Demolition Campaign in Chaurpati Rural Municipality of Achham district, Nepal.

### Methods

This was a cross-sectional quantitative study; conducted among 385 resident girls and women of the Chaurpati Rural Municipality, Achham who were in the menstrual life span (menarche to pre- menopause). A multistage sampling technique was used to select respondents from the six wards of Chaurpati Rural Municipality. Data were collected through individual interview using KOBO toolbox. Data were analyzed using IBM's SPSS 21. Ethical approval was obtained from IRC of Pokhara University. Appropriate descriptive statistics such as mean/median and SD were applied.

**Data availability statement:** All relevant data are within the paper and its Supporting Information files.

**Funding:** This research was supported by the University Grants Commission (UGC), Nepal, through a Faculty Research Grant awarded to principal investigator K.J. The funder had no role in study design, data collection and analysis, decision to publish, or preparation of the manuscript.

**Competing interests:** The authors have declared that no competing interests exist.

## Results

Most of the participants (93.5%) reported that existing *Chhaugoths* in their communities had been demolished. Consequently, 85.5% now reside in separate rooms at home during menstruation. However, 11% still use *Chhaugoths,* often covertly. Participants reported improved living arrangements (30.8%); yet menstrual hygiene management (MHM) practices remained poor. While, 48.6% used sanitary pads, disposal practices remained suboptimal, with 29.1% burned pads and 15.1% dumped them directly in water sources posing environmental health risks.

## Conclusion

This study demonstrates progress in shifting women from hazardous menstrual seclusion to safer, in-house accommodations following the *Chhaugoth* demolition campaign. However, critical gaps persist in menstrual waste disposal and supportive infrastructure. To achieve sustainable menstrual dignity and equity, interventions must combine targeted behavior change communication with early, pre-menarche education, to foster generational shifts in practices and norms. Further, studies explaining the factors associated with successful implementation of interventions and their effectiveness constitute the future scope of the study.

## Introduction

Menstruation is a natural physiological process experienced by women and girls of reproductive age, which is often stigmatized and linked to social taboos, and harmful practices worldwide, especially in low and middle income countries (LMIC) of South Asia [1,2]. Menstrual hygiene management (MHM) refers to the use of clean menstrual management materials, availability and access to soap and water, adequate privacy and dignity to manage and disposal of menstrual waste [3]. Despite increasing awareness of menstrual health as a human rights issue, over 500 million women and girls lack access to adequate menstrual hygiene materials. Menstruating women and girls face reproductive health risks, shame, embarrassment, school absenteeism and stigma [4–7].

In Nepal, many menstruating women experience and follow the practice of *Chhaupadi*, a form of menstrual exile in secluded huts known as *Chhaugoths* [8,9]. *Chhaupadi* is derived from two words: '*Chhau*' meaning 'menstruation' and '*padi*' meaning 'women' [10] where women are forced to isolate and sleep inside a small shed/hut known as *Chhaugoth* [8].This practice is particularly prevalent in Karnali and Sudurpaschim Provinces; including Achham district of Nepal [11,12].

Achham district has one of the highest number of *Chhaugoths*, in Nepal which is deeply influenced by these cultural norms and practices that dictates women and girls must isolate themselves during menstruation, with the belief that menstrual blood is impure and failing to do so will anger the deities and bring misfortune upon their families [1,13].

According to the documented reports, Women and girls face severe health consequences confined to *chhaugoths* such as increased risks of snakebites, wild animal attacks, suffocation, sexual violence, fire, hypothermia and even death [11,14,15]. Beyond physical harm, menstrual seclusion restricts access to nutrition, hygiene facilities and shelter contributing to reproductive health issues and social exclusion [16–18].Recent deaths of women and girls residing in *chhaugoth*s while adhering to this practice [19,20] underscore the urgent need for culturally sensitive public health interventions.

Recognizing menstrual health and hygiene as the fundamental right to health and human dignity, the Government and international organizations have actively advocated promoting safe menstrual health practices [21,22].

In May 2005, the Supreme Court of Nepal recognized *Chhaupadi* as a discriminatory and dangerous custom [8]. However, for many years *Chhaupadi* was decriminalized because of the absence of specific legislation until decades later change only came in 2017. In 2019, the Government of Nepal issued an eight-point circular to the local authorities and administration in 19 districts across Sudurpaschim and Karnali provinces with the active mobilization of local leaders, civil society and media for demolition of *Chhaugoths* [16,23,24]. The *Chhaugoth* demolition campaign was formally launched in late December 2019, following an intense implementation in early 2020 directed by the Ministry of Home Affairs to district administrations in 19 districts of Sudurpaschim and Karnali provinces [25,26] with campaign activities such as identification and mapping of the existing *chhaugoths* [27], awareness to the community and family members about the legal prohibition and health risks associated with this practice. In Achham district alone, over 10,000 *chhaugoths* were demolished with the involvement of police, local representatives, and community people [23,28].

Despite these bold interventions, the real impact on the ground remains unclear. There still lacks evidence regarding whether these efforts led to meaningful improvements in menstrual hygiene practices at community level. While some studies reported that criminalization of *Chhaupadi* tradition is perceived as a positive step for the behavior change, with a third of community people expressing their intent to abandon this practice after learning of legal penalties. Also, accurate awareness of the law remains low with the *chhaupadi* behavior change as a gradual as the social pressure to uphold the tradition persists [29].

Furthermore, the absence of sustained monitoring mechanism has led to mixed outcomes. Several media information reported that some women having no alternative shelter following demolition are forced to live in caves or hanging tarps leaving them even more vulnerable leading to unintended consequences [25,30]. In some areas, *chhaugoths* were rebuilt after campaigns ended as neither local nor police authorities prioritized the follow up [23]. Additionally, limited studies have examined the lived experiences of women following the demolition campaign; nonetheless, the realities whether legal reforms have created more dignified and safer menstrual health practice or not, which underscores the need of assessment of the campaign outcomes.

Although, healthy menstruation is increasingly recognized as a key indicator of overall health, dignity, and reproductive well-being, it remains entangled with harmful traditional practices such as *Chhaupadi*, a form of menstrual seclusion that requires girls and women to stay in *Chhaugoths* during their menstrual period in many rural areas of western Nepal. This study aims to assess the current living arrangements of menstruating girls and women and examine Menstrual Hygiene Management (MHM) practices in Chaurpati Rural Municipality of Achham.

## Materials and methods

### Study design and setting

This was a cross-sectional quantitative study conducted to assess menstrual hygiene management (MHM) practices of girls and women (participants) after *Chhaugoth* demolition campaign (2017 and subsequent anti-*chhaupadi* events). The study was conducted in Chaurpati Rural Municipality of Achham district, Nepal. This Municipality is administratively divided into 7 Wards and multiple sub-units called villages/tole.

### Study area selection

Chaurpati Rural Municipality was selected due to the reported ongoing prevalence of the *Chhaupadi* practice and its inclusion in the government's demolition campaigns (Ward number 7 was excluded due to the lack of evidence of the demolition campaign).

### Study population

The study population comprised of girls and women within the menstrual life span (menarche to pre menopause), who were residents of Chaurpati Rural Municipality. According to municipal records, the total population of girls and women aged 10–49 across the six included wards was 8466.

### Sample size determination

The sample size of 385 participants was determined using Slovin's formula with a 5% margin of error, assuming a prevalence of *Chhaupadi* practice (p = 50%) [31] with the 95% confidence level.

### Sampling procedure

A multistage sampling technique was used to select participants from six wards in Chaurpati Rural Municipality, Nepal. Fig 1 shows a flowchart of the selection process for this study.

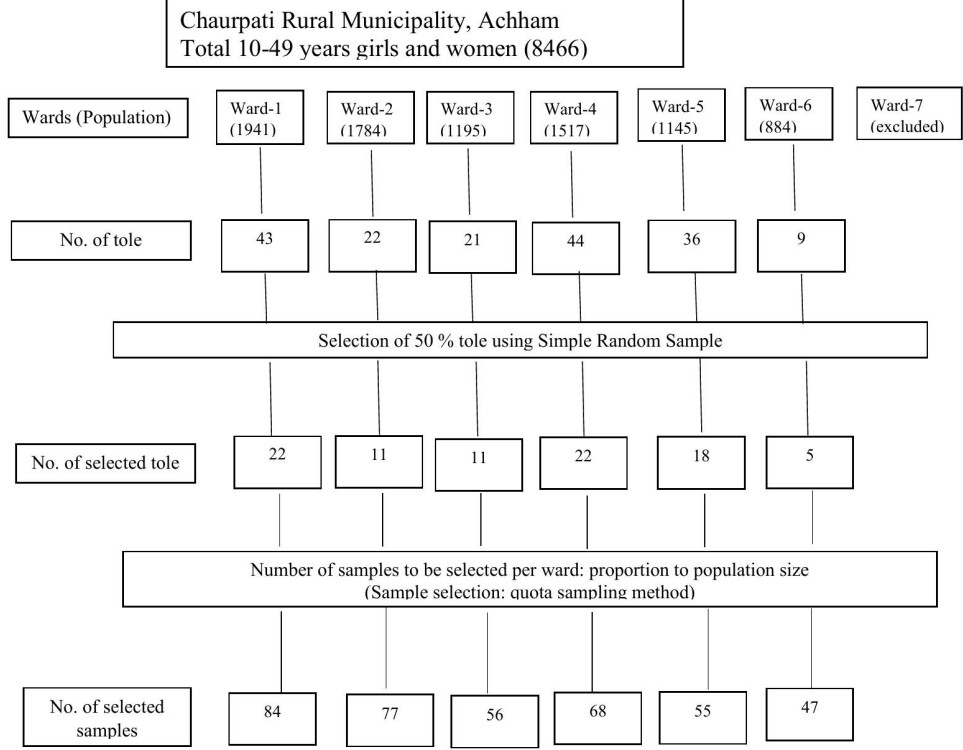

**Fig 1. Selection process.**

Stage 1: Of the seven wards in Chaurpati Rural Municipality, six wards (Wards 1–6) were included based on demolition campaign intervention. Ward 7 was excluded. An alphabetical listing of all included wards with villages/tole was prepared.

Stage 2: From each ward's alphabetical listing of all villages, 50% of villages were randomly selected

Stage 3: The total sample size of 385 participants was allocated proportionately across the six wards based on the population of girls and women aged 10–49 years in each ward. Within each selected village, quota sampling was used to select individual participants.

Stage 4: Enumerators visited household in the selected villages and approached eligible participants who provided the consent to participate were included until the required quota for that ward was achieved.

### Data collection

Data collection and participant recruitment took place from 26 May 2024–9 June 2024. Face to face interview was conducted with the participants to gather information on socio-demographic characteristics, living arrangements and MHM practices. Similarly, an observation checklist was used to assess the physical conditions of living place used during menstruation and/or to supplement the information. One day training was organized for the study team to collect and manage data using KOBO toolbox. Female enumerators having diploma or higher degree in nursing/ public health collected data with their android mobile phone. Tools were developed in both English and Nepali Language.

Trained enumerators collected data. A pretesting of the tool done among 10% sample participants of Sudurpaschim Province who were currently living in Pokhara metropolitan and/or studying in different institutes of Pokhara.

Collected data were exported to IBM SPSS 21 for analysis. Descriptive statistics (frequency, percentage, mean, median, minimum, maximum, standard deviation) were computed to describe the participant's sociodemographic characteristics and MHM practices. Current menstrual hygiene management practices were assessed of key practices such as the type of menstrual products, frequency of changing menstrual materials, cleaning and disposal practices, and personal hygiene behaviors.

### Ethical considerations

Ethical approval was obtained from the Institutional Review Committee (IRC) of Pokhara University (Ref no: 151/2080/2081); and additional approval was also obtained from the office of Chaurpati Rural Municipality. Written consent was obtained from all the participants prior to data collection. Written assent was obtained from participants under 18 years old and parental consent from their parents/guardians before their participation. Confidentiality and anonymity of the participants was maintained throughout the process.

## Results

### Socio demographic characteristics of participants

A total of 385 women/girls from six different wards were included this study, where 21.8% participants were from Ward no.1; followed by others (Table 1). Nearly half of the women/girls (42.9%) were aged 20–29 years (Mean age: 28.75±8.49 years). Majority of the respondents were Chhetri/Brahmin (70.7%), with nearly all of the women/girls were Hindus (98.7%). Regarding marital status, four out of five participants were married (80.5%); with more than half lived in Joint family (54.5%). Nearly two third participants were Housewife (64.4%) and the average monthly income was 107.17 USD, based on an exchange rate of Nepali rupees per 1 USD (Oct 3,2025). Additionally, more than a quarter of the participants had Secondary education (29.9%) and Informal Education (28.6%) and few had Bachelor's, Masters and higher degree (Table 1).

### Menstrual living arrangements before and after the campaign

More than three quarters of participants (75.8%) reported having ever stayed in a *Chhaugoth* during menstruation. Before *Chhaugoth* demolition, campaign, (74.8%) of participants regularly stayed in *Chhaugoth*, with majority staying for 4 days

**Table 1. Socio demographic characteristics of the respondents.**

| Sociodemographic characteristics | Frequency (n = 385) | Percentage |
|---|---|---|
| **Residence (Ward-wise)** | | |
| 1 | 84 | 21.8 |
| 2 | 77 | 20.0 |
| 3 | 56 | 14.5 |
| 4 | 66 | 17.1 |
| 5 | 55 | 14.3 |
| 6 | 47 | 12.2 |
| **Age (in completed years)** | | |
| Less than 20 | 59 | 15.3 |
| 20-29 | 165 | 42.9 |
| 30-39 | 104 | 27.0 |
| 40 or above | 57 | 14.8 |
| (Mean age: 28.75 ± 8.49 years; Min-Max: 13–49 years) | | |
| **Caste/Ethnicity** | | |
| Brahmin/Chhetri | 272 | 70.7 |
| Dalit | 103 | 26.8 |
| Janjati | 1 | 0.3 |
| Others (not disclosed) | 9 | 2.3 |
| **Religion** | | |
| Hindu | 380 | 98.7 |
| Christian | 5 | 1.3 |
| **Marital Status** | | |
| Married | 310 | 80.5 |
| Unmarried | 63 | 16.4 |
| Widow | 12 | 3.1 |
| **Family type** | | |
| Joint Family | 210 | 54.5 |
| Nuclear Family | 175 | 45.5 |
| **Occupation** | | |
| Housewife | 248 | 64.4 |
| Student | 58 | 15.1 |
| Farmer | 47 | 12.2 |
| Others (Daily wage worker, Laborer, Service holders) | 24 | 6.2 |
| Business | 8 | 2.1 |
| **Monthly Household Income (in USD):** Mean: 107.17 Median: 70.42 Range (Min-Max): 7.04–1056.56 **Note: (1 USD = 142 Nepalese rupees; exchange rate on date: Oct 3,2025) | | |
| **Education level** | | |
| Bachelor's level and above | 16 | 4.2 |
| Secondary level | 115 | 29.9 |
| Basic level | 86 | 22.3 |
| Informal Education | 110 | 28.6 |
| Illiterate | 58 | 15.1 |

#Brahmin/Chhetri refers to respondents from higher Hindu groups traditionally occupying priestly and warrior classes. Dalit includes respondents from marginalized groups. Janjati represents indigenous ethnic groups, while Others includes participants not classified under the above categories.

(35.1%) or 5 days (27.8%). More than a third of the participants used warm clothes (35.8%), followed by Local matrices (Moto Gadda) (25.5%), and few participants used straw materials or bare floor as a sleeping materials.

After the *Chhaugoth* demolition campaign, a great majority (85.5%) of the participants now stay in separate rooms within their homes during menstruation; however, 11.4% continue to use *Chhaugoths* and (3.1%) reported using other arrangements or did not disclose their location. Among those who still use *Chhaugoths*, nearly three quarters (72.7%) stay for 4 days. For the sleeping materials, (60.5%) now use warm clothes, thick local mattresses (36.6%) and only (0.3%) continue to sleep in bare floor during menstruation. Almost all the participants (93.5%) confirmed that their community *Chhaugoths* were demolished by the government's campaign. However, despite the demolition campaigns, (78.2%) reported that they still face restrictions during menstruation; indicating that the *Chhaupadi* tradition is still prevalent in the society (Table 2).

**Table 2. Menstrual Living Arrangements – Practices before and after Chhaugoth Demolition Campaign (n = 385).**

| Variables | Before Campaign | After Campaign |
| --- | --- | --- |
| **Ever Stayed in *Chhaugoth*#** | | |
| Yes | – | 292 (75.8) |
| No | – | 93 (24.2) |
| **Place of residence during menstruation** | | |
| *Chhaugoth* | 288 (74.8) | 44 (11.4) |
| Separate room in a house | 97 (25.2) | 329 (85.5) |
| Others (usual room, neighboring house/not disclosed) | 0(0.0) | 12 (3.1) |
| **Average duration of stay in Chhaugoth** | **(n = 385)** | **(n = 44)** |
| 4 days | 135 (35.1) | 32 (72.7) |
| 5 days | 107(27.8) | 9 (20.5) |
| 6 days | 46 (11.9) | 3 (6.8) |
| Others (no days specified)/no stay in Chhaugoth | 97 (25.2) | 0 (0.0) |
| **Type of sleeping materials used during menstruation** | | |
| Warm clothes | 138 (35.8) | 233 (60.5) |
| Local matrices (Motto Gadda)/simple matrices | 98 (25.5) | 10 (2.6) |
| Straw | 39 (10.1) | |
| Bare floor | 13(3.4) | 1 (0.3) |
| Others local products (old clothes, Jute Sacks/Burlap) | 97 (25.2) | 0(0.0) |
| Thick local mattresses | 0 (0.) | 141 (36.6) |
| **Practices/or restrictions followed during menstruation after the *Chhaugoth* demolition campaign##** | | |
| Yes | – | 301 (78.2) |
| No | – | 84 (21.8) |
| **Government's initiative to demolish *Chhaugoths* result in the demolition of interviewee Chhaugoth as well###** | | |
| Yes | NA | 360 (93.5) |
| No | NA | 25 (6.5) |

Note: Figures in the parenthesis indicate percentage of respective frequencies

# assessed only after the campaign ## no accurate data available but largely it was practiced.

### Not applicable

## Practices/restrictions included: not touching others, not entering kitchen/temple, food/ dietary restrictions, sleeping separately

### Menstrual hygiene management practices

**Menstrual products used.** Nearly half of the participants (48.6%) used sanitary pads. More than one-third (37.1%) of the participants continued to use old cloth rag, while a small minority used locally cloth pads, menstrual cups as menstrual hygiene products.

**Product accessibility and education.** Majority of the participants (61.6%) reported that these menstrual hygiene products were easily accessible, while 13.2% found them difficult to access and 5.2% reported no access at all. Over half of the participants (52.2%) had received education or training on safe Menstrual Hygiene Practices.

**Observed changes.** Half of the participants (50.6%) observed changes menstrual hygiene practices after the *Chhaugoth* demolition campaign. The observed changes were improved living conditions, improvements in their hygiene and sanitary conditions, and reduced stigma followed by use and disposal of pads and access to healthy diet (Table 3). Likewise, majority of the participants (87.8%) reported that they no longer stay in Chhaugoth after the elimination campaign.

**Frequency of changing materials.** (32.5%) changed their sanitary products as per need, followed by (26%) changed twice a day.

**Hygiene facilities and practices.** Among participants, (83.6%) reported that they always have access to clean water and soap, while 11.4% had occasional access and 4.9% had no access.

Bathing during menstruation was universal (99.7%), where 57.4% bathed on their first day, 23.4% on the second day and 16.9% on the fourth day. Additionally, nearly half of the respondents take their bath for one day (43.6%) and regularly for three days (41.6%) while only a small minority (14.8%) bathed regularly for two days after their first bath during menstruation.

Regarding genital hygiene practices, 99.5% clean their private parts during menstruation. Among these, 53% used soap and water, and 42.1% used water only.

**Toilet access during menstruation.** Majority (71.7%) used shared toilets at home during menstruation, while 25.5% did in open space, and only 2.6% had a separate toilet access specifically for menstrual use.

**Menstrual waste disposal practices.** The most common disposal practice was burning (29.1%), washing and reuse (25.6%), and 15.1% dumping directly into river stream. Small proportion of the respondents disposed menstrual waste in pits (8.3%), open area (1.8%), and through burial (0.3%).

**Menstrual communication.** Almost two thirds of the respondents (61.8%) felt comfortable sharing and discussing menstruation issues with others. Among those who shared, the most common confidants were peers (50%), family members (38.7%), school teachers (24.8%), and health workers (21.8%) (Table 3).

### Observational findings on living conditions

**Structural findings.** Majority (83.6%), of the participants had windows for ventilation in their living spaces, while16.4% stayed in places without proper ventilation. 90.4% had access to locks for privacy.

**Basic facilities.** Almost all the respondents (96.1%) had sleeping arrangements during menstruation. 73.8% had regular supply of food, while 26.2% faced food restriction during menstruation. 75.8% reported access to drinking water, leaving nearly a quarter of (24.2%) without adequate drinking water access. 88% had access to lighting, and electricity was accessed by 79%.

**Sanitation facilities.** Toilet was accessed by 68.8% respondents in their living arrangements. Among those with toilet access, only 9.4% had a separate toilet, while 88.3% shared toilets with their family.

**Waste management.** More than half of the respondents (56.4%) lacked proper disposal facilities for menstrual waste. Space for drying menstrual cloths was available to 88.1%.

**Location and proximity.** 57.9% stayed within 50 meters, 18.4% stayed within 50–100 meters away, and 23.7% stayed more than 100 meters from their house (Table 4).

**Table 3. Menstrual hygiene management related information (n = 385).**

| Variables | Frequency (n) | Percentage |
|---|---|---|
| **Menstrual Hygiene Product used** | | |
| Sanitary pad | 187 | 48.6 |
| Old cloth rags | 143 | 37.1 |
| Locally made cloth pad | 40 | 10.4 |
| Menstrual cups | 7 | 1.8 |
| Do not use anything/ Others (not disclosed) | 8 | 2.1 |
| **Accessibility of (Locally made cloth pad, Sanitary pads and menstrual cups) menstrual hygiene products** | | |
| Easily Accessible | 237 | 61.6 |
| Others (infrequent/ on demand available) | 77 | 20.0 |
| Difficult to Access | 51 | 13.2 |
| Not Accessible | 20 | 5.2 |
| **Received any education or Training on Safe Menstrual Hygiene Practices** | | |
| Yes | 201 | 52.2 |
| No | 184 | 47.8 |
| **Observed any changes in menstrual hygiene practices in the community after the *Chhaugoth* demolition campaign** | | |
| Yes | 195 | 50.6 |
| No | 190 | 49.4 |
| **If yes, Items of changes observed (n = 195)** | | |
| Improved living conditions (Safe housing, separate rooms, no longer in sheds/stables) | 60 | 30.8 |
| Hygiene & Sanitation (Daily bathing, cleanliness practices, toilet use) | 50 | 25.6 |
| Reduced Stigma | 33 | 16.9 |
| Menstrual Hygiene Practice (Pad Use, Use of sanitary pads, proper disposal methods) | 20 | 10.3 |
| Health and Safety (Protection from diseases/ snakes/cold, fewer infections, well-ventilated room) | 16 | 8.2 |
| Nutrition (Access to milk, yogurt, adequate food) | 9 | 4.6 |
| Others (self-respect, empowerment) | 7 | 3.5 |
| **Frequency sanitary products change during menstruation period** | | |
| As per need | 125 | 32.5 |
| Twice a day | 100 | 26.0 |
| Thrice a day | 55 | 14.3 |
| More than three times a day | 51 | 13.2 |
| Once a day | 33 | 8.5 |
| Others (not specified, cannot recall) | 21 | 5.5 |
| **Access to clean water and soap for hygiene during menstruation (n = 385)** | | |
| Always accessible | 322 | 83.6 |
| Sometimes accessible | 44 | 11.4 |
| Not accessible | 19 | 4.9 |
| **Places for disposal of used menstrual waste** | | |
| Burning | 112 | 29.1 |
| Others (wherever at ease in different places) | 76 | 19.8 |
| Washing and reuse | 99 | 25.6 |

*(Continued)*

**Table 3.** (Continued)

| Variables | Frequency (n) | Percentage |
|---|---|---|
| Dumping in river/stream | 58 | 15.1 |
| Pit | 32 | 8.3 |
| Open jungle/area dumping | 7 | 1.8 |
| Burying | 1 | 0.3 |
| **Practices of sharing menstruation-related problems with others (n=385)** | | |
| Yes | 238 | 61.8 |
| No | 147 | 38.2 |
| **If yes, with whom do you share the problems (n=238)- Multiple responses** | | |
| Peer group | 119 | 50.0 |
| Family members | 92 | 38.7 |
| School teacher | 59 | 24.8 |
| Health workers | 52 | 21.8 |
| Others (relatives, religious person) | 24 | 10.1 |
| **Bathing during menstruation (n=385)** | | |
| **Bathing during menstruation** | | |
| Yes | 384 | 99.7 |
| No | 1 | 0.3 |
| **If yes, on which days do you bathe during menstruation (n=384)** | | |
| First day | 221 | 57.4 |
| Second day | 90 | 23.4 |
| Fourth day | 65 | 16.9 |
| Third day | 8 | 2.1 |
| **Duration of bathing after first bath during menstruation (n=384)** | | |
| For one day | 168 | 43.6 |
| Regularly for three days | 160 | 41.6 |
| Regularly for two days | 57 | 14.8 |
| **Places for defecation and urination during menstruation (n=385)** | | |
| Shared toilet at home | 276 | 71.7 |
| Open space | 98 | 25.5 |
| Separate toilet at home | 10 | 2.6 |
| Others (other's toilet/public toilet/office toilet) | 1 | 0.3 |
| **Genitalia cleaning practices during menstruation (n=385)** | | |
| Yes | 383 | 99.5 |
| No | 2 | 0.5 |
| **Materials used to clean genital area during menstruation (n=383)** | | |
| Soap and water | 204 | 53.0 |
| Only water | 162 | 42.1 |
| Other (local herbs, samphoo) | 12 | 3.1 |
| Plain paper | 7 | 1.8 |

## Discussion

This study assessed the living arrangements and menstrual hygiene management (MHM) practices of participants following the government led *Chhaugoth* demolition campaigns in Chaurpati Rural Municipality, Achham district of Nepal. These findings reveal substantial progress in the living arrangements and hygiene behaviors.

**Table 4. Findings of observation.**

| Observation items | Response | Frequency (n) | Percentage (%) |
|---|---|---|---|
| Availability of ventilation/windows in living spaces | Yes | 322 | 83.6 |
| Availability of locks on doors and windows | Yes | 348 | 90.4 |
| Presence of place to dispose of used pads | Yes | 168 | 43.6 |
| Availability of sleeping arrangement in the living space | Yes | 370 | 96.1 |
| Availability of regular food in living space | Yes | 284 | 73.8 |
| Availability of drinking water in living space | Yes | 292 | 75.8 |
| Lighting arrangement inside living space | Yes | 339 | 88 |
| Availability of electricity in living space | Yes | 304 | 79 |
| Availability of toilet in living space | Yes | 265 | 68.8 |
| If yes, is there a separate toilet specifically for use during menstruation (n = 304) | Yes | 25 | 9.4 |
| | No | 234 | 88.3 |
| | Yes but others also use | 6 | 2.3 |
| Distance of the living space from nearby houses | < 50 meters | 223 | 57.9 |
| | 50–100 meters | 71 | 18.4 |
| | > 100 meters | 91 | 23.7 |
| Availability of space to dry menstrual clothes | Yes | 339 | 88.1 |

This study reported a remarkable decline in the Chhaugoth use, where a majority of the respondents (93.5%) reported the demolition of their *Chhaugoths* after the campaign. The decline in *Chhaugoth* use from 74.8% before to 11.4% after the demolition leading to significant improvement in the living arrangements which suggests behavioral shift and reflect the appreciable outcomes of the demolition campaign, where women and girls are protected and feel safe moving from isolated and hazardous sheds to within their homes.

Despite physical demolition of *Chhaugoths,* 78.2% of the respondents still face cultural taboos and restrictions during their menstrual period and are bounded to follow the seclusion practices within their home. This findings aligns with the studies from the previous studies from different areas of Nepal [1,32,33]; where women are secluded in livestock sheds or any specific rooms, which confirms the existence of the practice becoming a silent but impactful cultural restriction.

Furthermore, observational data indicates an increasing refinement but not a complete improvement, where still 16.4% of the respondents lacked proper ventilation menstrual waste disposal (56.4%), and consistent supply to safe and clean drinking water (24.2%) and food (26.2%). While compared to dreadful conditions of the traditional *Chhaugoths* reported by the study of Amatya [1] which indicated a slight improvement but this findings highlights that the current alternate seclusion rooms are not universally safe nor dignified.

Regarding MHM practices, 48.6% of respondents reported the use of sanitary pads which in comparison to other studies that reported much reliance to cloths/rags as an absorbent materials [1,32]. This increasing change could be due to the public awareness about reproductive health problems caused by unsanitary hygiene behaviors related to menstrual hygiene practice. Likewise, majority of the respondents feel comfortable in sharing menstruation related problems with others primarily to the peers (50%) and family members (38.7%). This finding aligns with other studies where their first source of information to menstruation is from their friends and family specifically their mother or sisters [6,34]. These findings signify a cultural shift towards openness and normalization of menstruation without feeling shame, where this silence often prevents women from seeking health support during menstruation. Despite such progress, unsafe and unreliable menstrual waste disposal practices remain unchanged. In this study, burning (29.1%) and directly dumping in rivers

(15.1%) are the common methods to disposal which is environmentally harmful and can pose multiple health risks. This findings is consistent to other studies of Kanchanpur [35] and Pyuthan [34] districts of Nepal, where the disposal are similar to the current practices of Achhami women and girls. These similarities in findings are mostly due to the fact that these are geographically and culturally connected areas of Sudurpaschim province suggesting regional norms and attributed to these regions shared cultural traditions regarding menstrual impurity and significant migration flows within the province.

This descriptive, exploratory study has several limitations. First, it's cross sectional nature limits the representation of the relationship between *Chhaugoth* demolition campaign and the observed changes. Second, the absence of pre-campaign data on Menstrual Hygiene Management practices for this population prevented a before after comparison. Third, the limited number of independent variables may not fully capture the determinants of Menstrual Hygiene Management outcomes. Fourth, given the criminalized status of *Chhaupadi,* social desirability bias may have led to underreporting of continued practice. Fifth,the study was conducted in a rural municipality which limits its generalizability. Finally, the exclusion of ward no. 7 due to the lack of information related to *Chhaugoth* demolition campaign could resulted in some sorts of selection bias.

## Conclusion

This study demonstrates remarkable progress in shifting from unsafe, unhygienic, hazardous menstrual seclusion to safer, in-house accommodations; reflecting positive changes following successful *Chhaugoth* demolition interventions. Living arrangements have improved with more girls and women now residing in secure and separate rooms during menstruation. However, critical gaps remain in menstrual hygiene management (MHM) practices, particularly regarding unsafe menstrual waste disposal and inadequate menstrual friendly infrastructures. This continues to impede safe and dignified MHM. There is the need to strengthen community awareness through targeted Behavior Change Communication (BCC) with the active involvement of family members to eliminate cultural taboos and norms and promote supportive environment. Furthermore, integrating menstrual health education before menarche, equipping girls with early knowledge, awareness and confidence to adopt safe hygiene practices. In addition, these early targeted interventions can foster generational change enabling future generation girls and women to achieve menstrual dignity and equity. We recommend that future research with comprehensive variables to identify factors associated with successful implementation of interventions and their effectiveness.

## Supporting information

**S1. Questionnaire.**
(PDF)

**S2. Inclusivity in global research questionnaire.**
(DOCX)

**S3. MHM SPSS Practices.**
(SAV)

## Acknowledgments

Authors are thankful to the authorities of Chaurpati Rural Municipality, Achham, Nepal; the local Police staff and the study participants, for their invaluable cooperation. The authors also expresses gratitude to the University Grant Commission (UGC), Nepal for funding and support to this study.

## Author contributions

**Conceptualization:** Kalpana Jnawali, Prativa Tiwari, Jagat Prasad Upadhyay, Sirjana Pandit Pahari.
**Data curation:** Kalpana Jnawali, Ayusha Ghale, Anushka Shrestha.

**Formal analysis:** Prativa Tiwari, Ayusha Ghale, Sirjana Pandit Pahari, Damaru Prasad Paneru.

**Funding acquisition:** Kalpana Jnawali, Jagat Prasad Upadhyay, Anjana Sigdel, Sapana Thapa, Kiran K.C..

**Investigation:** Kalpana Jnawali, Jagat Prasad Upadhyay, Anjana Sigdel, Sapana Thapa, Anushka Shrestha, Sirjana Pandit Pahari.

**Methodology:** Kalpana Jnawali, Ayusha Ghale.

**Project administration:** Kalpana Jnawali, Prativa Tiwari, Jagat Prasad Upadhyay, Anushka Shrestha.

**Resources:** Kalpana Jnawali, Prativa Tiwari, Ayusha Ghale, Jagat Prasad Upadhyay, Anushka Shrestha, Kiran K.C., Sirjana Pandit Pahari, Damaru Prasad Paneru.

**Software:** Prativa Tiwari, Ayusha Ghale, Kiran K.C., Damaru Prasad Paneru.

**Supervision:** Kalpana Jnawali, Prativa Tiwari, Damaru Prasad Paneru.

**Validation:** Kalpana Jnawali, Prativa Tiwari.

**Visualization:** Kalpana Jnawali, Prativa Tiwari, Ayusha Ghale, Anushka Shrestha.

**Writing – original draft:** Kalpana Jnawali, Ayusha Ghale, Anushka Shrestha.

**Writing – review & editing:** Prativa Tiwari, Jagat Prasad Upadhyay, Sirjana Pandit Pahari, Damaru Prasad Paneru.

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
