## [Decision Letter · Decision Letter 0]

8 Feb 2026

PONE-D-25-57219Menstrual Hygiene Management Practices among Girls and Women after the Chhaugoth Demolition Campaign in Chaurpati Rural Municipality, Achham, NepalPLOS One

Dear Dr. Ghale,

Thank you for submitting your manuscript to PLOS ONE. After careful consideration, we feel that it has merit but does not fully meet PLOS ONE’s publication criteria as it currently stands. Therefore, we invite you to submit a revised version of the manuscript that addresses the points raised during the review process. ===============================Four reviewers value the topic of this manuscript but suggest a number of changes. ===============================

We look forward to receiving your revised manuscript.

Kind regards,

Alison Parker

Academic Editor

PLOS One

Journal Requirements:

Reviewers' comments:

Reviewer's Responses to Questions

**Comments to the Author**

1. Is the manuscript technically sound, and do the data support the conclusions?

Reviewer #1: Yes

Reviewer #2: Yes

Reviewer #3: Partly

Reviewer #4: Partly

2. Has the statistical analysis been performed appropriately and rigorously? 

Reviewer #1: Yes

Reviewer #2: No

Reviewer #3: Yes

Reviewer #4: No

3. Have the authors made all data underlying the findings in their manuscript fully available?

Reviewer #1: No

Reviewer #2: Yes

Reviewer #3: Yes

Reviewer #4: No

4. Is the manuscript presented in an intelligible fashion and written in standard English?

Reviewer #1: Yes

Reviewer #2: Yes

Reviewer #3: Yes

Reviewer #4: No

5. Review Comments to the Author

Reviewer #1: Menstrual Hygiene Management Practices among Girls and Women after the Chhaugoth Demolition Campaign in Chaurpati Rural Municipality, Achham, Nepal

The study investigates menstrual hygiene management practices among women who practice Chhaupadi, a tradition in which women stay in a hut during menstruation in western Nepal. The study contributes to the emerging literature on the intersection of women’s health, menstrual hygiene practices, and government policies. The study also has important findings, for example, that the government’s hut demolition policy has been effective: the majority of women have abandoned the practice of staying in huts during menstruation. However, the study finds that women's hygiene practices remained poor. The method section of the study is also strong: it is a well-designed cross-sectional survey with a modest sample size. While the paper is well thought out and well written, it could be stronger if the following comments are addressed.

Here are my comments:

1. Although the study’s objective is to focus on menstrual hygiene management, one of the important findings from the study is that the government intervention of demolishing the huts had worked. 85% of women and girls reported abandoning and significantly changing the practice after the demolition campaign. The authors could make this finding more prominent. I would recommend providing more information about the demolition campaign in the communities—how the demolition was conducted, who was involved, how the community reacted, and how the government enforced it. And slightly reframe the paper as an evaluation of the policy. This could make the paper stronger.

2. While this is an excellent descriptive study, it could also benefit from adding additional statistical analysis. A simple t-test to determine whether the differences in responses are significant could be added to the descriptive tables. Furthermore, I would encourage authors to consider how demographics, such as age/education/caste, are associated with outcomes such as hygiene practice. You could do this by using linear regression models. Here, you could use menstrual hygiene practices as your outcome variable, such as burning menstrual waste. The objective would be to see whether there are differences in menstrual hygiene practices by age or education or ethnicity. It may provide important insights on targeted policy making for local government. Here is one example of such an analysis, in which authors investigated the association between Chhaupadi and children’s health.

a. Women’s extreme seclusion during menstruation and children’s health in Nepal S Joshi, Y Acharya - PLOS Global Public Health, 2022

Minor comments:

1. Title is too long. I would suggest re-titling it.

2. In page 19: sentence 21, the authors have written “Menstruation, also called Chhaupadi”. Menstruation is not called Chhaupadi. Menstruation is called “Chhau” and Chhaupadi is the woman having her menses.

3. In page 19: sentence 21: The word extricable is not necessary. Delete it.

4. Italicize the Nepali words such as Chhaupadi, Chhaugoth.

5. Page 4-sentence 56: rewrite the first sentence. It is clunky.

6. Page 4-sentence 68: Chhaupadi tradition is not rooted in Hindu belief; it is rooted in the traditional Masto-worshipping beliefs among the Khas people.

7. Page 7-sentence 129: Can you provide a visual chart for the design? For example: how many villages were selected in the first stage? Which stage was randomized? How many (exact numbers) were selected in the next stage? Did you randomize for wards or villages? Who were the interviewers? How were they trained?

8. Page 12: last line: “Practices/or restrictions followed during menstruation after the Chhaugoth demolition campaign”: And the response is yes and no. What does this mean? It is unclear.

9. Rather than having everything under one table titled Results, having multiple subtitles with sub-themes will add clarity and flow. The subtitle could be descriptive results, hygiene-related results, observation results, etc. Although this study is descriptive, you could also write it more analytically and have a flow with some interesting narratives. The result section currently reads like a report, with little consideration of why you are providing us with the data.

10. Currently, the tables are not up to par for the publication for this journal. I strongly suggest editing the tables; maybe looking at some of the examples from the published paper would help.

Reviewer #2: This paper addresses an important and timely topic. Menstrual practices in Nepal are under-researched, and the manuscript makes a valuable contribution by documenting practices, beliefs, and constraints that have meaningful implications for health, education, and gender equity. The use of descriptive statistics is appropriate and necessary for establishing the scope and prevalence of these practices, and these results form an important foundation for the analysis.

That said, the rigor and interpretability of the findings could be strengthened through the use of more advanced statistical modeling. While descriptive statistics effectively summarize patterns in the data, they do not fully account for confounding factors or heterogeneity across individuals and contexts. Incorporating multivariable regression or other inferential models would allow the authors to better isolate associations between key predictors (e.g., socioeconomic status, education, geography, caste/ethnicity) and menstrual practices, thereby strengthening causal interpretation and policy relevance.

In addition, more advanced modeling would enable the authors to formally test hypotheses suggested by the descriptive results and assess the robustness of observed relationships. This would help distinguish which factors are independently associated with menstrual practices versus those that may be correlated due to underlying structural or demographic differences. Such analyses would enhance confidence in the findings and clarify which determinants may be most actionable for intervention or program design.

Overall, the manuscript presents important descriptive evidence on a critical topic. With the addition of more rigorous statistical analysis, the paper would make an even stronger contribution to the literature and provide clearer guidance for researchers, practitioners, and policymakers working in this area.

Reviewer #3: Strength:

- Strong justification of the research problem.

- Clear linkage between menstruation stigma, chhaupadi and health risks.

Comments:

- Please arrange the keywords in alphabetical order.

- Some ideas are repeated multiple times in introduction section: Menstruation stigma, Legal prohibition of Chhaupadi, Health risk associated with Chhaugoths.

- Use 'menstrual hygiene management (MHM) consistently after first definition.

- In methods section: Solvin's formula - should be Slovin's formula.

- How quotas were determined at household/ individual level were not clearly explained.

- Spelling mistake like maters for masters

- In many tables, others were not explained clearly in result section

- Majority of respondent were repeated excessively in result section

- In discussion section, some paragraphs mix living conditions, cultural restrictions and waste disposal.

Reviewer #4: Thank you for the opportunity to review this timely manuscript examining menstrual hygiene management (MHM) practices in Achham, Nepal after the chhaugoth demolition campaign. The authors are to be commended for engaging with a culturally rich and complex topic with significant implications for adolescent wellbeing and community health. Empirical evidence emerging from interventions, such as chhaugoth demolition campaigns, is especially valuable for advancing the field, and the focus on post-intervention practices has the potential to meaningfully inform future programs and policy efforts.

Below, I provide a series of comments and suggestions aimed at strengthening the clarity, methodological rigor, and contextual depth of the manuscript. Overall, there are several key components of the paper that I would suggest significantly expanding/reworking to be ready for publication, and hence suggest a major revision.

PLEASE SEE ATTACHMENT FOR DETAILED COMMENTS.

6. PLOS authors have the option to publish the peer review history of their article (what does this mean?). If published, this will include your full peer review and any attached files.

Reviewer #1: No

Reviewer #2: No

Reviewer #3: **Yes:** Isha Amatya

Reviewer #4: No

---

## [Author Response · Author response to Decision Letter 1]

5 Apr 2026

Title: Menstrual Hygiene Management Practices after the Chhaugoth Demolition Campaign in Achham, Nepal

Dear reviewers, Thank you for the opportunity to revise our manuscript. We greatly appreciate the thoughtful and constructive feedback provided by the reviewers, which has sustainably improve the quality of our paper. According to the reviewer’s feedback, we have carefully addressed and edited each comments and suggestion.

#Reviewer 1:

1. Although the study’s objective is to focus on menstrual hygiene management, one of the important findings from the study is that the government intervention of demolishing the huts had worked. 85% of women and girls reported abandoning and significantly changing the practice after the demolition campaign. The authors could make this finding more prominent. I would recommend providing more information about the demolition campaign in the communities—how the demolition was conducted, who was involved, how the community reacted, and how the government enforced it. And slightly reframe the paper as an evaluation of the policy. This could make the paper stronger.

Response: We appreciate this constructive feedback. In response, we have expanded our introduction detailing information about the demolition campaign, including how it was conducted, who was involved, and community responses (line 92-99). We have also reframed and modified in accordance with policy perspective in the Introduction.

2. While this is an excellent descriptive study, it could also benefit from adding additional statistical analysis. A simple t-test to determine whether the differences in responses are significant could be added to the descriptive tables. Furthermore, I would encourage authors to consider how demographics, such as age/education/caste, are associated with outcomes such as hygiene practice. You could do this by using linear regression models. Here, you could use menstrual hygiene practices as your outcome variable, such as burning menstrual waste. The objective would be to see whether there are differences in menstrual hygiene practices by age or education or ethnicity. It may provide important insights on targeted policy making for local government. Here is one example of such an analysis, in which authors investigated the association between Chhaupadi and children’s health.

a. Women’s extreme seclusion during menstruation and children’s health in Nepal S Joshi, Y Acharya - PLOS Global Public Health, 2022

Response: We thank the reviewer for this suggestion. We considered this study as descriptive one and with its exploratory nature. The scope of study was to assess the status of living arrangements and hygiene management practices after the wider impactful events. And we did not frame this in the form of policy effectiveness study. due to limitation in the pre-campaign measureable indicators and variables. The primary aim was to describe living arrangements and MHM practices following the demolition campaign. The limited number of independent variables and the absence of pre-campaign data for this population prevented a before-after comparison. Therefore, as noted in the Limitations section (Lines 321-329), we restricted our analysis to descriptive statistics to accurately reflect the study's scope. We acknowledge that more rigorous studies with comprehensive variables are needed to identify factors associated with campaign implementation success, as stated in the Conclusion and recommendations.

3. Title is too long.

Response: We have shortened the title to "Menstrual Hygiene Management Practices after the Chhaugoth Demolition Campaign in Achham, Nepal" (Line 1).

4. In page 19: sentence 21, the authors have written "Menstruation, also called Chhaupadi". Menstruation is not called Chhaupadi. Menstruation is called "Chhau" and Chhaupadi is the woman having her menses.

Response: We have corrected this throughout. In the Abstract (Lines 22-24), we now state: "Menstruation, also called as Chhau... Women who menstruate known as Chhaupadi." In the Introduction (Lines 68-69), we define Chhaupadi as derived from Chhau (menstruation) and padi (woman).

5. In page 19: sentence 21: The word extricable is not necessary.

Response: We have removed "extricable" and replaced it with "inextricably" in the Abstract (Line 23).

6. Italicize the Nepali words such as Chhaupadi, Chhaugoth.

Response: All Nepali terms (Chhau, Chhaupadi, Chhaugoth) are now italicized throughout the manuscript.

7. Page 4-sentence 56: rewrite the first sentence. It is clunky.

Response: We have rewritten the first sentence of the Introduction (Lines 58-60).

8. Page 4-sentence 68: Chhaupadi tradition is not rooted in Hindu belief; it is rooted in the traditional Masto-worshipping beliefs among the Khas people.

Response: We have corrected this cultural attribution in the Introduction (Lines 70-72).

9. Page 7-sentence 129: Can you provide a visual chart for the design? For example: how many villages were selected in the first stage? Which stage was randomized? How many (exact numbers) were selected in the next stage? Did you randomize for wards or villages? Who were the interviewers? How were they trained?

Response: We have provided a detailed flowchart (Figure 1) showing the multistage sampling procedure, including exact numbers of population per ward, random selection of 50% of villages, proportional allocation of samples, and quota sampling. We have also described enumerator training in the Methods section (Lines 164-165), noting that female enumerators with diploma or higher degrees in nursing/public health were trained on KOBO data collection.

10. Page 12: last line: "Practices/or restrictions followed during menstruation after the Chhaugoth demolition campaign": And the response is yes and no. What does this mean? It is unclear.

Response: We have clarified this in Table 2 with a detailed footnote (Line 219-220) specifying that restrictions included not touching others, not entering the kitchen or temple, food and dietary restrictions, and sleeping separately.

11. Rather than having everything under one table titled Results, having multiple subtitles with sub-themes will add clarity and flow.

Response: We have reorganized the Results section with clear thematic subheadings (Lines 183-280): "Socio demographic Characteristics of Participants," "Menstrual Living Arrangements Before and After the Campaign," "Menstrual Hygiene Management Practices," "Observational Findings on Living Conditions."

12. Currently, the tables are not up to par for the publication for this journal.

Response: We have reformatted all tables to meet PLOS ONE standards.

#Reviewer 2:

1. The rigor and interpretability of the findings could be strengthened through the use of more advanced statistical modeling. Incorporating multivariable regression or other inferential models would allow the authors to better isolate associations between key predictors and menstrual practices, thereby strengthening causal interpretation and policy relevance.

In addition, more advanced modeling would enable the authors to formally test hypotheses suggested by the descriptive results and assess the robustness of observed relationships. This would help distinguish which factors are independently associated with menstrual practices versus those that may be correlated due to underlying structural or demographic differences. Such analyses would enhance confidence in the findings and clarify which determinants may be most actionable for intervention or program design.

Response: We thank reviewer 2 for the constructive suggestions, and we appreciate this. However, our study is descriptive and exploratory in nature, designed as a status assessment of living arrangements and MHM practices following the demolition campaign rather than a policy effectiveness study. As noted in the Methods section (Lines 172-175) and Limitations (Lines 321-329), the scope is limited to describing current practices. The limited number of independent variables and the absence of pre-campaign data for this population prevented a before-after comparison. We have explicitly justified this in the Limitations section and clarified the exploratory nature of the analysis. We acknowledge that more rigorous studies with comprehensive variables are needed to identify factors associated with campaign implementation success.

#Reviewer 3:

1. Please arrange the keywords in alphabetical order.

Response: We have reordered the keywords alphabetically (Line 54): "Chhaugoth Demolition Campaign, Chaurpati Rural Municipality, Menstrual Hygiene Management Practice, Nepal."

2. Some ideas are repeated multiple times in introduction section: Menstruation stigma, Legal prohibition of Chhaupadi, Health risk associated with Chhaugoths.

Response: We have streamlined the Introduction, consolidating repeated ideas into cohesive paragraphs and removing redundant statements. These changes are reflected throughout the Introduction section (Lines 58-120).

3. Use 'menstrual hygiene management (MHM) consistently after first definition.

Response: We have defined MHM at first mention in the Introduction and used the acronym consistently throughout the manuscript.

4. In methods section: Solvin's formula -should be Slovin's formula.

Response: We have corrected this to "Slovin's formula" in the Methods section (Line 139).

5. How quotas were determined at household/ individual level were not clearly explained.

Response: We have added a detailed explanation in the sampling procedure (Stage 4, Lines 152-154). Every household with girls and women was considered as the unit of study because household practices related to living arrangements may be similar, resulting in homogeneous practice of following Chhaupadi tradition.

6. Spelling mistake like maters for masters.

Response: We have corrected "maters" to "masters" in Table 1.

7. In many tables, others were not explained clearly in result section.

Response: We have added explanatory footnotes for all "others" categories in Tables 1-4.

8. Majority of respondent were repeated excessively in result section.

Response: We have varied the language throughout the Results section, using alternatives such as "most participants," "nearly half," "more than one-third," and "a small minority”, or simply using their percentages.

9. In discussion section, some paragraphs mix living conditions, cultural restrictions and waste disposal.

Response: We have reorganized the Discussion section paragraph (Lines 286-320).

#Reviewer 4:

Abstract Comments:

Comment: Please exercise caution in how the term chhaupadi is introduced. Ensure consistency between abstract and introduction.

Response: We have revised the abstract to define Chhaupadi accurately, consistent with the Introduction (Lines 68-69).

Comment: Grammar and word choice throughout the abstract requires improvement.

Response: We have thoroughly edited the abstract for grammar and clarity.

Comment: Please specify the year(s) in which the chhaugoth demolition campaign was implemented and clarify timing.

Response: We have specified that the campaign was launched in late December 2019 with implementation intensifying in early 2020 (Line 93), and that data were collected from 26 May to 9 June 2024 (Lines 159).

Comment: Clarify participant sampling by specifying whether participants were selected from all wards. Consider replacing “different wards” with a more explicit description.

Response: We have clarified in the Abstract (Line 34) and Methods (Line 142) that participants were selected from six wards (wards 1-6), with Ward 7 excluded due to lack of campaign evidence.

Comment: Grammar throughout the Results portion of the abstract needs substantial revision, as it is currently difficult to follow.

Response: Thank you for your constructive feedback. We have revised our result section.

Introduction Comments:

Comment: Writing and grammar throughout the manuscript need careful revision to improve clarity and flow.

Response: We have carefully edited the entire manuscript for grammar and flow.

Comment: It is not sufficiently clear what the chhaugoth demolition campaign entailed. Provide additional detail on who, what, duration, scope, and community engagement.

Response: We have provided extensive detail in the Introduction (Lines 87-99) as described in response to Reviewer 1.

Comment: Lines 109–112, which describe the broader health implications of chhaupadi, may be better positioned earlier.

Response: We have moved health implications earlier in the Introduction (Lines 77-83) to frame public health relevance.

Comment: Critically, the introduction does not adequately address the complexities and unintended consequences of shed demolition and the chhaugoth campaign. Prior research has documented that some women report having no alternative shelter following demolition. This potential harm—and community concerns around it—should be acknowledged upfront to provide a more balanced framing of the intervention.

Response: We have added a critical discussion of unintended consequences in the Introduction (Lines 107-114), documenting that after demolition, some women had no alternative shelter and were forced to live in caves or under hanging tarpaulins.

Methods Comments:

Comment: Line 121 refers to "subsequent events"; please clarify what these events were.

Response: Subsequent events refers to the events following after chhaugoth demolition campaign (anti chhaupadi events).

Comment: The authors note that the study area was selected due to the reported prevalence of chhaupadi and inclusion in government demolition campaigns. However, other municipalities likely met these criteria. Please explain how this specific municipality was selected (e.g., government recommendation, logistical considerations).

Response: We have explained in the Study Area Selection (Lines 130) that Chaurpati was selected due to reported ongoing prevalence of Chhaupadi and its inclusion in government demolition campaigns, following government recommendations.

Comment: The assumed chhaupadi prevalence of 50% used for sample size estimation requires citation.

Response: We have added a citation to Kothari (2004) for Slovin's formula (Line 140).

Comment: Please describe enumerator training in greater detail, including duration, content, and any steps taken to ensure data quality. Additionally, provide information on enumerators’ gender and other relevant positionality characteristics.

Response: We have added that female enumerators with diploma or higher degrees in nursing/public health collected data, and one-day training was organized for the study team on KOBO data collection (Lines 164).

Comment: Lines 144–146 describe data collected on MHM practices. Given that chhaupadi practices are also a central focus of the paper, please clarify what questions were asked regarding isolation practices (e.g., sleeping location, restrictions, duration) and consider providing example questions.

Response: We have clarified this in the Methods and added a footnote in Table 2 specifying restrictions assessed and attached questionnaire as a supplementary file.

Comment: Please review and revise language related to consent procedures for participants under 18 years of age; typically, both assent and parental/guardian consent are required.

Response: We have revised to state: "Written assent was obtained from participants under 18 years old and parental consent from their parents/guardians before their participation" (Line 180-181).

Comment: The current analysis relies primarily on descriptive statistics, which provides a useful overview of participant characteristics and MHM practices. However, the analytic approach could be strengthened by incorporating inferential statistical tests to examine associations between participant characteristics and key behaviors.

Specifically, the authors may consider conducting bivariate analyses to assess whether sociodemographic factors (e.g., age, education, marital status, household income, or location) are associated with outcomes such as menstrual product use, frequency of changing materials, hygiene and disposal practices, or sleeping in a chhaugoth. Chi-square tests could be used for categorical variables, with t-t

---

## [Decision Letter · Decision Letter 1]

17 May 2026

Menstrual Hygiene Management Practices after the Chhaugoth Demolition Campaign in Achham, Nepal

PONE-D-25-57219R1

Dear Dr. Ghale,

We’re pleased to inform you that your manuscript has been judged scientifically suitable for publication and will be formally accepted for publication once it meets all outstanding technical requirements.

Kind regards,

Alison Parker

Academic Editor

PLOS One

Additional Editor Comments (optional):

Reviewers' comments:

Reviewer's Responses to Questions

**Comments to the Author**

1. If the authors have adequately addressed your comments raised in a previous round of review and you feel that this manuscript is now acceptable for publication, you may indicate that here to bypass the “Comments to the Author” section, enter your conflict of interest statement in the “Confidential to Editor” section, and submit your "Accept" recommendation.

Reviewer #1: All comments have been addressed

Reviewer #3: All comments have been addressed

2. Is the manuscript technically sound, and do the data support the conclusions?

Reviewer #1: Yes

Reviewer #3: Yes

3. Has the statistical analysis been performed appropriately and rigorously? 

Reviewer #1: Yes

Reviewer #3: Yes

4. Have the authors made all data underlying the findings in their manuscript fully available?

Reviewer #1: (No Response)

Reviewer #3: Yes

5. Is the manuscript presented in an intelligible fashion and written in standard English?

Reviewer #1: Yes

Reviewer #3: Yes

6. Review Comments to the Author

Reviewer #1: (No Response)

Reviewer #3: Thank you for your response. All the previous comments has been addressed. I have no further comments.

7. PLOS authors have the option to publish the peer review history of their article (what does this mean?). If published, this will include your full peer review and any attached files.

Reviewer #1: No

Reviewer #3: **Yes:** Isha Amatya

---

## [Editor Report · Acceptance letter]

PONE-D-25-57219R1

PLOS One

Dear Dr. Ghale,

I'm pleased to inform you that your manuscript has been deemed suitable for publication in PLOS One. Congratulations! Your manuscript is now being handed over to our production team.

Kind regards,

on behalf of

Dr. Alison Parker

Academic Editor

PLOS One